# Lipid-Lowering Effects of a Novel Polysaccharide Obtained from Fuzhuan Brick Tea In Vitro

**DOI:** 10.3390/foods12183428

**Published:** 2023-09-14

**Authors:** Wenjuan Yang, Shirui Cheng, Meng Liu, Nan Li, Jing Wang, Wenbo Yao, Fuxin Chen, Jianwu Xie, Pin Gong

**Affiliations:** 1School of Food Science and Engineering, Shaanxi University of Science and Technology, Xi’an 710021, China; yangwenjuan@sust.edu.cn (W.Y.); 220412099@sust.edu.cn (S.C.); liumeng102412@126.com (M.L.); linan@sust.edu.cn (N.L.); wangjingsp@sust.edu.cn (J.W.); yaowenbo@sust.edu.cn (W.Y.); xiejw@sust.edu.cn (J.X.); 2School of Biological and Pharmaceutical Sciences, Shaanxi University of Science and Technology, Xi’an 710021, China; 3School of Chemistry and Chemical Engineering, Xi’an University of Science and Technology, Xi’an 710054, China; chenfuxin1981@163.com

**Keywords:** Fuzhuan brick tea polysaccharides, lipid lowering, AMPK/SREBP-1c/FAS pathway

## Abstract

Lipid accumulation causes diseases such as obesity and abnormal lipid metabolism, thus impairing human health. Tea polysaccharide is one of the natural, active substances that can lower lipid levels. In this paper, an oleic-acid-induced HepG2 cell model was established. The lipid-lowering effects of a novel group of Fuzhuan brick tea polysaccharides (FTPs)—obtained from Fuzhuan brick tea—were examined in vitro. The monosaccharide composition of FTP3 was Glc, Gal, Ara, Man, Rha, GalAc, GlcAc, and Xyl with a molar ratio of 23.5:13.2:9.0:5.5:5.4:2.7:1.3:1.0, respectively. A molecular weight of 335.68 kDa was identified for FTP3. HepG2 cells treated with FTP3 achieved a prominent lipid-lowering effect compared with cells treated with oleic acid. Images of the Oil Red O staining treatment showed that FTP3-treated groups had significantly fewer red fat droplets. TC and TG levels were lower in FTP3-treated groups. FTP3 alleviated lipid accumulation in HepG2 cells, activated AMPK, and decreased the SREBP-1C and FAS protein expressions associated with fatty acid synthesis. FTP3 holds promising potential for its lipid-lowering effects.

## 1. Introduction

At present, with increasing consumption levels, poor living habits such as staying up late, lack of exercise, and irregular diet are on the rise, which often results in increased lipid levels in the human body. Excessive lipid accumulation in the body causes lipid metabolism disorders, which then lead to obesity, nonalcoholic steatohepatitis, and other metabolic diseases [1]. Excessive levels of non-esterified fatty acids and carbohydrates, as well as oxidative damage, are the major causes of metabolic diseases [2]. Currently, reducing such lipid accumulation is an important method of preventing or delaying the formation of lipid metabolism disorders. Although many drugs are currently used to treat lipid metabolism disorders, adverse reactions can damage human health to a certain degree; reported adverse effects include liver damage, gastrointestinal dysfunction, kidney damage, and other diseases [3]. Against this background, a substance with the potential to decrease lipid accumulation and antioxidant levels is urgently needed.

Natural chemical compounds derived from plants are called phytochemicals, which have recently received increasing attention because of their substantial pharmacological activities and few side effects [4,5]. Fuzhuan brick tea (FBT) is a class of dark tea made from the leaves of a tea plant (*Camellia sinensis*) that has undergone fungal fermentation [6]. FBT is considered an ethnobotanical medicine of the Uygur, Tibetan, and Mongolian ethnic groups, inhabiting the southwestern and northwestern border regions of China. These peoples traditionally consume a diet high in meat and milk but low in vegetables and fruits. The consumption of FBT for an extended time has been shown to reduce lipid and grease levels and has healthcare and pathological prevention effects on the human body [7]. FBT contains various kinds of bioactive ingredients such as tea polyphenols and polysaccharides [8]. In particular, Fuzhuan brick tea polysaccharide (FTP) is widely valued for its hypoglycemic and lipid-lowering effects [9]. Chen et al. confirmed that FTP is a typical acidic heteropolysaccharide that significantly alleviates oxidative damage in mice on a high-fat diet [10]. Related disease research has illustrated that FBT has weight-loss and lipid-lowering effects in vivo [11,12]. FBT has been shown to significantly inhibit the gain of body weight and accumulation of adipose tissue [13]. Moreover, FBT can activate pancreatic amylase and protease, downregulate serum total protein (TP) levels, reduce serum alanine transaminase and aspartate transaminase activities, and decrease serum triacylglycerol (TG) and total cholesterol (TC) levels [14]. Xiao et al. showed that the aqueous extract of FBT at a concentration of 500 μg/mL has potent lipid-lowering activity in a high-fat zebrafish model [15]. However, the mechanism underlying the lipid-lowering effect of FTP remains unclear. In this study, FTP was isolated from FBT, and its ability to inhibit lipid accumulation in HepG2 cells induced by oleic acid was examined.

## 2. Materials and Methods

### 2.1. Materials

FBT was purchased from Shaanxi Jingwei Fuzhuan Brick Tea Co., Ltd. (Shaanxi Jingwei Fuzhuan Brick Tea Co., Ltd., Xianyang, China). DEAE-cellulose DEAE-52 (3 cm × 12 cm, 0.45 μm), cellulase, papain, α-glucosidase (≥700,000 U/mL), acarbos (95%), PNPD, α-amylase (50 U/mg), Sephadex G-200 (Φ20 × 600 mm), and microporous membranes were obtained from Shanghai Yuanye Biotechnology Co., Ltd. (Yuanye Biotechnology Co., Ltd., Shanghai, China). DMEM high-glucose medium, penicillin–streptomycin, and trypsin-EDTA were obtained from Bioengineering Co., Ltd. (Bioengineering Co., Ltd., Shanghai, China). Oil Red O staining solution and blue cell viability dye were obtained from Suolaibao Biotechnology Co., Ltd. (Suolaibao Biotechnology Co., Ltd., Beijing, China). Anti-β-actin, anti-SREBP-lc, anti-FAS, anti-AMPK, and anti-P-AMPK were obtained from Cell Signaling Technology (Cell Signaling Technology, Boston, MA, USA). All other reagents were of analytical grade and were obtained from Tianli Chemical Reagent Co., Ltd. (Tianli Chemical Reagent Co., Ltd., Tianjin, China).

### 2.2. Isolation and Purification of Polysaccharides from FBT

A mixture of papain and cellulase (2:1, *w*/*w*) at a concentration of 2% was added to 50 g of FBT powder and extracted for 60 min at pH 5.0 and 55 °C. The supernatant of the extract was filtered and concentrated. Protein was removed with Sevage reagent (n-butanol:chloroform = 1:4). The polysaccharide was refrigerated at 4 °C for 12 h and then centrifuged, fully dissolved in water, and freeze-dried. The obtained polysaccharide powder was then dissolved in distilled water, and the DEAE-cellulose DEAE-52 column chromatography method was used for purification. Purification was followed by an elution using NaCl (0.3 M, 0.5 mL/min). The section was collected and then purified with distilled water through Sephadex G-200 columns. The major peaks were collected, lyophilized, and named FTP3 [16].

### 2.3. Structural Characterization of FTP3

#### 2.3.1. Polysaccharide, Protein, Uronic Acid, and Total Phenolic Contents

The polysaccharide content was determined in reference to the phenol-sulfuric acid method. The protein content was determined with the Bradford method, and the total phenol content was measured with the Folin–Ciocalteu method [17,18].

#### 2.3.2. Ultraviolet Spectrum Scan and Fourier-Transform Infrared (FT-IR) Spectrum Analysis

UV spectral analysis was conducted according to a previously published method [19], with slight modification. The scanning range was 200–400 nm using 10 nm intervals. The infrared spectrum of FTP3 was measured in an infrared region of 4000–400 cm^−1^ using an FT-IR spectrometer (VERTE70, Bruker, Karlsruhe, Germany) [20].

#### 2.3.3. Scanning Electron Microscopy (SEM) Analysis

In brief, using a capillary tube, FTP3 powder was immersed in conductive gel and mounted on a sample holder. After gold sputtering, SEM (SU3500, Hitachi, Tokyo, Japan) was conducted to examine and image the surface structure of the polysaccharide sample [21].

#### 2.3.4. Molecular Weight (Mw) and Monosaccharide Composition

In accordance with the above-mentioned method, high-performance size-exclusion chromatography coupled with multi-angle laser light scattering and a refractive index detector (HPSEC-MALLS-RID, Wyatt Technology Co., Santa Barbara, CA, USA) was used to determine the Mw of FTP3 [22]. FTP3 separation was measured at 45 °C using a Shodex OHpak SB-806 M HQ (300 mm × 8.0 mm) column. The monosaccharide composition of FTP3 was determined using high-performance liquid chromatography (HPLC). A Phenomenex Gemini C18 110A (150 mm × 4.6 mm, 5 μm) column was used for the HPLC (U3000, ThermoFisher, Waltham, MA, USA) with a Phenomenex Gemini C18 110A (150 mm × 4.6 mm, 5 μm) column [23].

#### 2.3.5. Nuclear Magnetic Resonance (NMR) Spectroscopy Analysis

FTP3 (20 mg) was dissolved in heavy water. Then, the sample was filtered through a microporous filter membrane (0.22 μm) and subsequently transferred to an NMR tube for 1H-NMR and 13C-NMR analysis (AVANCEIII HD, Saarbrucken, Germany) [24].

#### 2.3.6. Congo Red Analysis

For the following series of experiments, the FTP3 solution was prepared at a mass concentration of 0.5 mg/mL; Congo red solution was prepared at a concentration of 50 μM; and NaOH solution was prepared at a concentration of 0–0.50 M. Congo red maximum absorption wavelengths were measured at 200–600 nm in different concentrations of NaOH solution.

#### 2.3.7. Atomic Force Microscope (AFM) Analysis

With continuous stirring at 25 °C for 4 h, FTP3 (10 μg) was dissolved in ultrapure water (1 mL). Samples were dropped onto a mica carrier, dried at 70 °C under ambient pressure, and analyzed [25].

### 2.4. In Vitro Antioxidant and Glucolipid Ameliorating Activity Assay of FTP3

#### 2.4.1. Antioxidant Activity

FTP3 and vitamin C (Vc) were dissolved in distilled water to obtain solutions with concentrations of 0.025, 0.05, 0.1, 0.2, 0.4, and 0.8 mg/mL. Vc was used as a positive control. To each of the above concentration gradients (2 mL), DPPH solution (0.05 mg/mL) was added, and the mixtures were vortexed (A_2_). The DPPH solution was replaced by absolute ethanol, and the mixture was vortexed (A_1_). Absolute ethanol was used instead of the sample solution, and the mixture was vortexed (A_0_). After the solution was in the dark at 25 °C for 30 min, the absorbance was determined at 517 nm. The clearance rate of DPPH· in the sample was calculated according to Equation (1) [26]:(1)$$\%DPPH\cdot radical\;scanvenging=(1-\frac{A_{2}-A_{1}}{A_{0}})\times 100\%$$

A total of 1 mL of each concentration of FTP3 and 1 mL each of 9 mmol/L H_2_O_2_, 9 mmol/L FeSO_4_, 9 mmol/L salicylic acid absolute ethanol solution, and distilled water were mixed, and the mixture was vortexed and incubated at 37 °C for 35 min. The absorbance value at 510 nm was measured (A_2_). The H_2_O_2_ solution was replaced with distilled water (A_1_), and the sample was replaced with distilled water (A_0_). The clearance rate of ·OH in the sample was calculated according to Equation (2) [27]:(2)$$\%OH\cdot radical\;scanvenging=(1-\frac{A_{2}-A_{1}}{A_{0}})\times 100\%$$

To 1 mL of the test solution, 3 mL Tris-HCl (pH 8.2) was added, and the mixture was incubated at 30 °C for 20 min. Then, 3 mL of 5 mmol/L pyrogallic acid was added at 25 °C, mixed, and left to settle for 3 min; then, the reaction was terminated by the addition of 1 mL of HCL. A_2_ represents the absorbance of FTP3 measured at 320 nm. A_1_ and A_0_ represent the absorbance of distilled water instead of the pyrogallic acid solution and distilled water instead of the sample solution in the blank group at 320 nm, respectively. The clearance rate of O_2_^−^· in the sample was calculated according to Equation (3) [28]:(3)$$\%O_{2}-radical\;scanvenging=(1-\frac{A_{2}-A_{1}}{A_{0}})\times 100\%$$

#### 2.4.2. Inhibition of α-Glucosidase and α-Amylase Activity

The inhibitory activity of FTP3 on α-glucosidase was measured based on the method of Li et al. [29] and Ren et al. [30], with minor modifications. The α-amylase inhibitory activity was assayed based on the method of Cao et al. [31] and Wang et al. [32], with minor modifications.

#### 2.4.3. Binding Rates of Cholesterol and Bile Acids

Cholesterol micelle solution was prepared (the final solution contained cholesterol and oleic acid at 5 mmol/L; lecithin and sodium taurocholate at 10 mmol/L; sodium chloride at 132 mmol/L; and phosphate-buffered saline (PBS) at 15 mmol/L at a pH of 7.4). FTP3 (10 mg) was added to micelle solutions, and micelle solution without a sample was used as a blank control. Samples were centrifuged after shaking at 37 °C, the supernatant was obtained, and the cholesterol level was measured with a kit (Nanjing Jiancheng Bio Co., Ltd., Nanjing, China). The cholesterol solubility of the sample group is C_1_, and the cholesterol solubility of the blank group is C_0_. The cholesterol binding rate was calculated as (1 − C_1_/C_0_) × 100%.

Mother liquor of sodium taurocholate, sodium glycocholate, and sodium cholate solution (0.3 mmol/L) was prepared with 0.1 mol/L of PBS at pH 7.6. Seven tubes with plugs were filled with 0, 0.1, 0.5, 1.0, 1.5, 2.0, and 2.5 mL of the above mother liquor. Then, PBS was added to bring the total amount to 2.5 mL, and 7.5 mL of sulfuric acid solution (60%) was also added. After 20 min of incubation in a water bath at 70 ℃, the solution was cooled in ice water for 5 min and measured at 387 nm. After this, FTP3 was put into these test tubes; then, hydrochloric acid solution (0.01 mol/L, 1 mL) was added, and the mixture was shaken and digested at 37 °C for 2 h. The pH was adjusted to 7.6, 4 mL of cholate solution was added, and the mixture was oscillated at 37 °C for 2 h. The absorbance of FTP3 was measured, and the bile acid salt concentration was obtained based on the standard curve. The concentration of bile acid salt in the sample solution is C_1_, and the concentration in the blank solution is C_0_. The bile acid binding rate was calculated as (1 − C_1_/C_0_) × 100% [32].

#### 2.4.4. The Survival Rate of HepG2 Cells with FTP3 and OA Treatment

To examine the mechanisms by which FTP3 improves lipid metabolism, an in vitro cell model of the human hepatoma cell line (HepG2) was used to study the disruption of lipid metabolism. HepG2 cells were obtained from Cell Bank, Chinese Academy of Sciences. The cells were grown in DMEM containing 2% (*v*/*v*) heat-inactivated fetal bovine serum (FBS), and a penicillin–streptomycin mixture was used to incubate cells in 5% CO_2_ at 37 °C. All experiments used cells at the logarithmic stage [33].

Cells were washed with PBS and digested with trypsin. The appropriate complete medium was added, and the cell suspension was centrifuged at 1000 rpm for 3 min. Then, the supernatant was discarded, and a complete medium was added. Cell suspension was added to a 96-well plate, and the concentration of cells was maintained at 1 × 10^4^ per well. Cells were cultured with FTP3 at concentrations of 0, 12.5, 25, 50, 100, 200, and 400 μg/mL prepared in serum-free medium for 24 and 48 h. Cells were cultured with oleic acid (OA) solutions (prepared with medium containing 2% FBS) at concentrations of 0, 100, 200, 300, 400, and 500 μM for both 24 and 48 h. After incubation, 100 μL of 5% MTT solution was added to each well in the dark. After 4 h of incubation in 5% CO_2_ at 37 °C, the supernatant was aspirated out, and 100 μL of DMSO was added to each well. The viability of cells was calculated by shaking the 96-well plate for 10 min and measuring the absorbance at 490 nm.

#### 2.4.5. Oil Red O Staining and Lipid Droplet Content

The preparation method of the stock solution is described in the following: OA was dissolved in 0.1 mol/L NaOH and heated in a water bath at 70 °C to obtain a 100 mmol/L stock solution. The solution was slowly added to BSA solution with a mass fraction of 10% in a water bath (55 °C) to prepare OA solution at a concentration of 10 mmol/L; this solution was then filtered through a 0.22 μm filter membrane to remove bacteria, and the filtrate was frozen at −20 °C. Before use, the cells were bathed in water at 55 °C for 15 min, cooled down to 25 °C, and diluted to the desired concentration with DMEM high-glucose medium containing 2% FBS. Intracellular lipid droplets were detected using Oil Red O staining with a concentration of 400 μM. Following this staining step, the unbound dye was removed by rinsing the cells with water. Stained lipid droplets were visualized within cells using a digital camera attached to a light microscope at a 200-fold magnification. Microplate readers at 490 nm were used to measure cholesterol accumulation [34].

#### 2.4.6. Determination of Biochemical Indicators Related to Lipid Metabolism and Antioxidants in HepG2 Cells

Cells were lysed using lysis buffer and centrifuged at 4 °C for 15 min at 4000 rpm. In the cell supernatant, the levels of TC, TG, and malondialdehyde (MDA) were measured. The activities of alanine transaminase (ALT), aspartate aminotransferase (AST), superoxide dismutase (SOD), glutathione peroxidase (GPX), and reactive oxygen species (ROS) were assessed using commercial kits (Nanjing Jiancheng Bioengineering Institute, Nanjing, China) according to the instructions of the manufacturer [35].

#### 2.4.7. Protein and Gene Expression Related to Lipid Metabolism in HepG2 Cells

Following the instructions of the manufacturer, total RNA was extracted using a high-purity total RNA rapid extraction kit (Nanjing Jiancheng Bio Co., Ltd., Nanjing, China), and the absorbances of the RNA extracted from the sample were measured at 260–280 nm [36]. Using All-in-One cDNA Synthesis SuperMix for qPCR, RNA samples with purity ratios (A260/A280) in the range of 1.8–2.0 were synthesized into single-strand cDNA. A list of real-time quantitative PCR primers used in this study is provided in Table 1.

Western blot was performed following a previously reported method [37]. As primary antibodies, SREBP-lc (1:3500), AMPK, anti-AMPK (Tr172), β-actin, and FAS antibodies (1:1000) were used, which were all purchased from Cell Signaling Technology (Beverly, MA, USA). Secondary rabbit antibody (1:1000) was purchased from Biyuntian Biological Reagent Co., Ltd. (Shanghai, China).

### 2.5. Statistical Analysis

Data obtained from triplicate experiments were calculated as means ± SD. and differences between groups were assessed via one-way analysis of variance. The statistical analyses were conducted using SPSS version 17.0. Differences were considered statistically significant at *p* < 0.05.

## 3. Results

### 3.1. Structural Characterization of FTP3

#### 3.1.1. Polysaccharide, Protein, Uronic Acid, and Total Phenol Contents

The polysaccharide content of FTP3 was 77.72 ± 0.83%, the content of protein was 0.28 ± 0.15%, the total phenol content was 1.63 ± 0.22%, and the uronic acid content was 20.09 ± 0.62% (identifying FTP3 as an acidic polysaccharide).

#### 3.1.2. Spectral Analysis

In a scanning range of 200–400 nm, there was no absorption peak in FTP3 after DEAE-cellulose and Sephadex G-200 column chromatography. This lack of absorption peak indicated that the method successfully removed proteins from samples and thus obtained protein-free FTP3 (Figure 1A).

The FT-IR spectrum of FTP3 is shown in Figure 1B. Characteristic peaks of hydrogen-bonded O-H stretching vibrations were found at 3394 cm^−1^, and C-H stretching vibrations were found at 2956 cm^−1^. The broad bands at 1650–1900 cm^−1^ were attributed to C=O stretching vibrations. The signals at 1597 cm^−1^ and 1412 cm^−1^ were attributed to uronic acid stretching vibrations, which identified FTP3 as an acidic polysaccharide [10]. The presence of pyranose was indicated by three absorption peaks between 1000 cm^−1^ and 1200 cm^−1^. The absorption of FTP3 was around 873 cm^−1^, caused by a linkage of β-glycosides. As shown by the absorption band at 818 cm^−1^, FTP3 had a linkage of α-glycosides in its molecular structure.

#### 3.1.3. Mw and Monosaccharide Composition

As shown in Figure 1C, the Mw of FTP3 was 335.68 kDa, as determined using gel permeation chromatography. The monosaccharide composition of FTP3 was Glc, Gal, Ara, Man, Rha, GalAc, GlcAc, and Xyl, and the molar ratio was 23.5:13.2:9.0:5.5:5.4:2.7:1.3:1.0, respectively (Figure 1D).

#### 3.1.4. Congo Red Analysis

Figure 1E shows the FTP3 and Congo red maximum absorption wavelengths in 0–0.5 mol/L NaOH solution. These results illustrate that the maximum absorption wavelength of FTP3 did not change significantly with an increasing concentration of NaOH, indicating that there was no triple-helix structure.

#### 3.1.5. NMR Analysis

As shown in Figure 1F, both chemical shifts of the anomeric hydrogen of FTP3 were greater than 5.0 ppm and lower than 5.0 ppm. Therefore, it can be inferred that the polysaccharide contains both α-pyranose and β-pyranose, which is consistent with the FT-IR results. As shown in Figure 1G, there was no heterocycle signal between δ82 ppm and δ88 ppm in the FTP3 terminal carbon. This result further indicates that the sugar residues in FTP3 are pyran-type, δ95-101 ppm is an α configuration, and δ101-105 ppm is a β configuration, which is in agreement with the results of ^1^H-NMR, presented above. The δ > 170 ppm carbon signal indicates that FTP3 contains uronic acid. The chemical shift of FTP3 demonstrated signal peaks at 175.80 ppm and 182.55 ppm, indicating that FTP3 may contain uronic acid, which is an acidic polysaccharide.

#### 3.1.6. SEM and AFM Analyses

SEM images provide visual evidence of FTP3 morphology. As shown in Figure 1H, under 2000-fold magnification, the composition of FTP3 is complex and dense, showing a curly layered structure. Under 5000-fold magnification, FTP3 showed an irregular sheet-like structure with loose pores, which may be caused by the loose aggregation of polysaccharide chains.

The plan view and the cubic spectrum of the AFM of FTP3 are shown in Figure 1I. The planar spectrogram contains many spherical and irregular clumps. The cube plot demonstrates that the polysaccharide had an irregular hill-tip morphology, which may be due to the aggregation of polysaccharide molecules.

### 3.2. In Vitro Antioxidant Activity

Concentration-dependent DPPH radical scavenging activity was verified in 0.025–0.8 mg/mL FTP3. The highest scavenging activity was 84.82% of 0.8 mg/mL FTP3 (Figure 2A). With increasing concentration, with a concentration range of 0.025–0.8 mg/mL, FTP3 became more effective at scavenging hydroxyl radicals. The highest hydroxyl radical scavenging activity was 73.27% of FPT3 with a concentration of 0.8 mg/mL (Figure 2B). The highest superoxide anion radical scavenging activity was 74.85% of 0.8 mg/mL FPT3 (Figure 2C).

### 3.3. In Vitro Inhibition of α-Glucosidase and α-Amylase Activity

As a result of the use of α-glycosidase and α-amylase inhibitors, glucose levels from carbohydrates decreased in response to the inhibition of enzyme activities, thus delaying the absorption of carbohydrates by the human body. As shown in Figure 2D,E, all tested samples presented a dose-dependent inhibitory effect on α-amylase and α-glucosidase. In the α-glucosidase inhibition assay, the highest scavenging activity of FTP3 was 81.75% at a concentration of 0.8 mg/mL (Figure 2D). In the α-amylase inhibition assay, the highest scavenging activity of FTP3 was 77.24% at a concentration of 0.8 mg/mL (Figure 2E).

### 3.4. In Vitro Binding to Bile Acids and Cholesterol Activity

The in vitro binding capacity of FTP3 to bile acid salts was 53.86% (Figure 2F), which was 85.66% of the in vitro binding capacity of the simvastatin positive control. The in vitro cholesterol binding inhibition capacity of FTP3 was 52.34% (Figure 2G), which was 67.33% of the in vitro cholesterol binding inhibition capacity of the simvastatin positive control.

### 3.5. Effect of FTP3 on Lipids Metabolism in HepG2 Cells

#### 3.5.1. The Survival Rate of HepG2 Cells with FTP3 and OA Treatment

The effect of FTP3 on cell viability was detected using an MTT assay. As shown in Figure 3A,B, compared with the control group, the treatment of HepG2 cells with FTP3 below 100 μg/mL promoted a certain level of cell proliferation. HepG2 cell survival was inhibited when treated with 200 μg/mL and 400 μg/mL of FTP3. Therefore, for subsequent experiments, FTP3 concentrations of 25 μg/mL, 50 μg/mL, and 100 μg/mL were selected and labeled as low-, medium-, and high-dose groups, respectively. The effect of OA on the viability of HepG2 cells was determined using the MTT method, and the results are shown in Figure 3C,D. Compared with the control group, no significant inhibitory effect was observed when HepG2 cells were treated with OA below 400 μM. However, when HepG2 cells were treated with 500 μM OA, the survival rate of cells decreased. To exclude the effect of OA on cell activity, an OA concentration below 400 μM was selected for the next experiment.

#### 3.5.2. Oil Red O Staining and Lipid Droplet Content Assay

Six groups of HepG2 cells were tested: a negative control (NC) group, an oleic acid (OA) group (400 μM OA), a low-dose (LD) group (400 μM OA + 25 μg/mL FTP3), a medium-dose (MD) group (400 μM OA + 50 μg/mL FTP3), a high-dose (HD) group (400 μM OA + 100 μg/mL FTP3), and a positive control (PC) group (100 μg/mL simvastatin). As demonstrated in Figure 4, the supplementation of the culture medium with FTP3 significantly reduced intracellular lipid accumulation, and this effect was amplified with increasing FTP3 concentrations.

#### 3.5.3. Biochemical Indicators Related to Lipid Metabolism and Antioxidants in HepG2 Cells

TG and TC concentrations in HepG2 cells were determined to evaluate the role of FTP3 in lipid accumulation. TG and TC levels were decreased progressively following FTP3 treatment for 24 h over a sample range of 25, 50, and 100 μg/mL. The low, medium, and high doses of TG levels following FTP3 treatment were 1.66, 1.48, and 1.34 mmol/gprot, respectively (Figure 5A), and TC levels were 1.69, 1.42, and 1.13 mmol/gprot, respectively (Figure 5B). These results suggest that FTP3 can reduce lipid accumulation in hepatocytes.

The AST and ALT activities of hepatocytes are important liver injury indicators. In the experiments, following the OA treatment of HepG2 cells, the AST activity reached 11.34 U/gprot, and a noticeable difference was found between the group treated with OA and the control group in terms of ALT activity (59.12 U/gprot). The treatment of cells with different concentrations of FTP3 significantly reduced AST and ALT activities compared with the OA group. These results indicated that FTP3 could attenuate damage caused by the OA treatment in HepG2 cells (Figure 5C,D).

SOD and GPX can effectively scavenge free radicals and prevent oxidative damage. As depicted in Figure 5E,F, a significant decrease in antioxidase activities (SOD and GPX) was observed when HepG2 cells were treated with OA. However, the activities of the above antioxidant enzymes were significantly enhanced in HepG2 cells treated with FTP3 and increased with increasing concentrations of FTP3. According to all the data, a significant increase in antioxidant enzyme activity and a reduction in oxidative damage were observed after FTP3 treatment, indicating that FTP3 treatment enhanced antioxidase activities.

MDA levels provide an indication of oxidative damage in cells, as this indirectly reflects the severity of free radical attacks. As shown in Figure 5G,H, following the OA treatment of HepG2 cells, the MDA content and ROS activity in the OA group increased significantly to 23.76 nmol/mgprot and 39.12%, respectively. However, the activities of MDA and ROS in the HepG2 cells decreased significantly when treated with FTP3 and decreased further with increasing FTP3 concentrations. In conclusion, OA promoted the production of ROS in cells, while FTP3 treatment significantly inhibited ROS production.

### 3.6. Gene and Protein Expression Associated with Lipid Metabolism

As depicted in Figure 6A, OA reduced AMPK mRNA expression, while different concentrations of FTP3 significantly increased AMPK mRNA expression levels by 50.12%, 59.52%, and 63.31%. As shown in Figure 6B,C, the expression level of mRNA in adipogenesis-related genes (SREBP-1c and FAS) increased significantly following OA treatment. In contrast, different concentrations of FTP3 clearly decreased the mRNA levels of SREBP-1c and FAS. As depicted in Figure 6D,E, mRNAs for IL-6 and TNF-α were up-regulated following OA treatment at 400 μM, while FTP3 counteracted these OA-induced increases.

FTP3 affected the expression of AMPK, p-AMPK, SREBP-1c, and FAS in HepG2 cells. As indicated in Figure 6F, there was a decrease in AMPK phosphorylation caused by OA, while its phosphorylation increased with increasing concentrations of FTP3. As shown in Figure 6G,H, SREBP-1c and FAS were downregulated following treatment with 400 μM OA, while FTP3 counteracted these OA-induced increases. These results suggest that FTP3 regulates the expression of AMPK, SREBP-1c, and FAS to exert beneficial metabolic effects.

## 4. Discussion

In this study, the novel polysaccharide FTP3 was obtained via isolation and purification from FBT. FTP3, which has a molecular weight of 335.68 kDa, is composed of Rha, Ara, Gal, Glc, Xyl, Man, GalAc, and GlcAc, identifying it as an acidic polysaccharide. Excessive levels of free radicals can lead to increased oxidative stress (OS) in vivo. OS is associated with the occurrence and development of a variety of diseases arising from disorders of lipid metabolism [38]. Plant polysaccharides can scavenge excessive free radicals in cells and show antioxidant activity [39]. Studies have shown that Qimen County Black Tea can scavenge DPPH·, ·OH, and O_2_^−^· [40], and the antioxidant activity of polysaccharides is directly proportional to the content of uronic acid in the tea [41]. The in vitro antioxidant effect of FTP3 extracted from FBT is consistent with what was found by Sun et al. [40], showing that FBT affected the scavenging activity on DPPH·, ·OH, and O_2_^−^·, which may be due to the presence of uronic acid.

Zhu et al. [42] demonstrated that the hypoglycemic activity of dark tea polysaccharides is related to their ability to inhibit α-glucosidase and α-amylase activities. This inhibition ability was positively correlated with the contents of uronic acid and certain monosaccharides, including arabinose, galactose, galactose, mannose, and uronic acid. This experiment identified the inhibitory effect of FTP3 on α-glucosidase and α-amylase, providing data support for its hypoglycemic activity. The excessive intake of lipids can cause metabolic diseases such as fatty liver and endanger people’s health [43]. Combined with bile acids, polysaccharides can promote the metabolism of cholesterol in the liver, thus achieving lower blood lipid levels. Research has shown that plant-hydrophobic polysaccharides have a strong affinity for cholesterol and can adsorb cholate [44]. The results of this study are consistent with these results for cholesterol inhibition and provide empirical support for subsequent lipid-lowering experiments on cells. OA is a plentiful fatty acid in the human diet and can cause lipotoxicity [45]. Therefore, to form a lipid accumulation model, in this study, OA was applied to damage HepG2 cells, and the role of FTP3 in regulating lipid metabolism was explored. Research has shown that lipid metabolism disorders are closely related to oxidative stress and inflammation, and these mechanisms interact with each other [46]. It has also been reported that the most intuitive indicators of lipid metabolism disorders are TC and TG contents; elevated TC and TG levels may be indicative of intracellular fat accumulation and increased risk of NAFLD [47,48]. A certain quantity of ROS is produced in cells undergoing oxidative stress, which can lead to abnormal lipid metabolism and inflammation [49]. The results of this study indicate that the treatment of OA-induced HepG2 cells with FTP3 resulted in a decrease in intracellular TC and TG contents and a decrease in the viabilities of ALT and AST. These decreases demonstrate that FTP3 can improve lipid deposition and alleviate liver function impairment. After FTP3 treatment, SOD and GPX activities increased, MDA and ROS contents decreased, and the mRNA expressions of IL-6 and TNF-α also decreased. These results showed that FTP3 can ameliorate lipid deposition, oxidative damage, and inflammatory symptoms induced by OA in HepG2 cells.

By activating the AMPK pathway, cells can adjust lipid metabolism by influencing genes and protein expressions associated with lipid deposition [50]. The SREBP-1c and FAS genes play important roles in the synthesis of fatty acids, and SREBP-1c can regulate the expression of FAS genes. The activation of AMPK has been shown to reduce adipogenesis in hepatocytes by downregulating the expression of SREBP-1c [51]. AMPK can inhibit SREBP-1c expression and reduce the activity of enzymes related to fatty acid synthesis, thus inhibiting lipid production, which can effectively treat diseases caused by abnormal lipid metabolism [52]. In this experiment, AMPK mRNA expression in HepG2 cells was upregulated with increasing FTP3 concentrations. This result is in accordance with the results of Jin et al. [53], who found that *Schisandra* polysaccharides adjust AMPK expression in a strict dose-dependent relationship. FTP3 may reduce lipid levels and promote fatty acid oxidation by activating AMPK. These results illustrated that the mRNA expressions of SREBP-1c and FAS were noticeably upregulated after OA-induced lipid deposition in HepG2 cells, which was in accordance with the observed increased lipid deposition in HepG2 cells. Compared with the OA group, both SREBP-1c and FAS were downregulated in the mRNA expression after FTP3 treatment, indicating that FTP3 can reduce lipid levels by inhibiting fatty acid synthesis. In the present study, the upregulation of P-AMPK protein expression and the downregulation of both SREBP-1c and FAS protein expressions induced by FTP3 indicated that, in HepG2 cells, OA-induced lipid accumulation was alleviated. FTP3 may exert lipid-lowering effects by regulating AMPK, SREBP-1c, and FAS protein expressions (Figure 7).

The polysaccharide structure is closely related to activity [54,55]. The composition of FTP3 is complex and dense, showing an irregular sheet-like shape with loose pores. It has been reported that loose and disordered structures may substantially reduce the diffusion rate of cholesterol, thus conveying a potential lipid-lowering effect that aids the treatment of metabolic diseases [56,57]. Significant antioxidant activities were reported for both α-configured pyranose and β-configured pyranose [58]. Xiao et al. [56] showed that plant polysaccharides containing Glc, Man, Xyl, Ara, and Gal can activate AMPK expression. Man, Gal, and GalA with uronic acid in the structure and monosaccharide composition can downregulate SREBP-1c and FAS mRNA and protein expressions in hepatocytes [59]. The results of the present study showed that FTP had apparent in vitro antioxidant and lipogenesis accumulation reduction activities; furthermore, FTP alleviated the oxidative damage caused by OA, proving that FTP can regulate lipid metabolism. It can be speculated that FTP3 had a positive effect on lowering lipid levels by regulating the protein expressions of AMPK, SREBP-1C, and FAS. Its effect may be related to α-pyranose, β-pyranose, and the monosaccharide composition of Glc, Man, Xyl, Ara, Gal, and GalAc. In future experiments, the mechanism underlying the effect of FTP3 on the AMPK/SREBP-1c/FAS pathway will be examined through protein inhibitors or agonists.

## 5. Conclusions

FTP3 is an acidic polysaccharide that has a powerful effect on decreasing TG, TC, ROS, and MDA contents and the activities of ALT and AST while also increasing the activities of SOD and GPX in HepG2 cells induced by OA. In Oil Red O-stained FTP3 samples, significantly fewer red fat droplets were observed. Moreover, FTP3 stimulated both the gene expression and protein expression of AMPK, SREBP-1C, and FAS. The current research used in vitro cell experiments and showed that FTP3 has a lipid-lowering effect. This is a preliminary study of FTP3. In the future, a high-fat animal model will be established, and the lipid-lowering effect and dosage of FTP3 will be determined in vivo. In addition, a simulated digestion method will be constructed in vitro to analyze the degree and rate of FTP3 digestion. This study provides a basis for the application of FTP3 to functional foods with the ability to regulate lipid accumulation.

## Figures and Tables

**Figure 1 foods-12-03428-f001:**
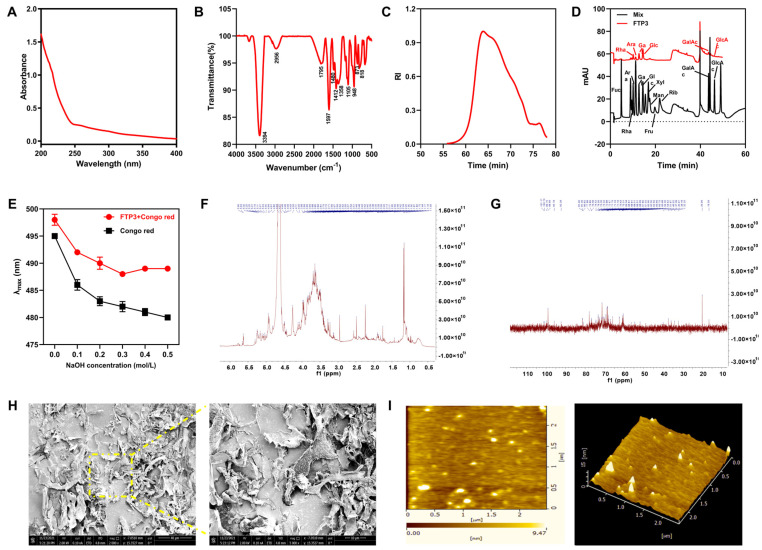
Structural characterization of FTP3. (**A**) UV spectrum. (**B**) FT-IR spectrum. (**C**) Absolute molecular weight analysis diagram. (**D**) HPLC chromatograms. (**E**) Congo red in NaOH solutions with different concentrations. NMR spectrum: (**F**) ^1^H-NMR, (**G**) ^13^C-NMR. (**H**) SEM images (2000-fold and 5000-fold). (**I**) AFM of FTP3.

**Figure 2 foods-12-03428-f002:**
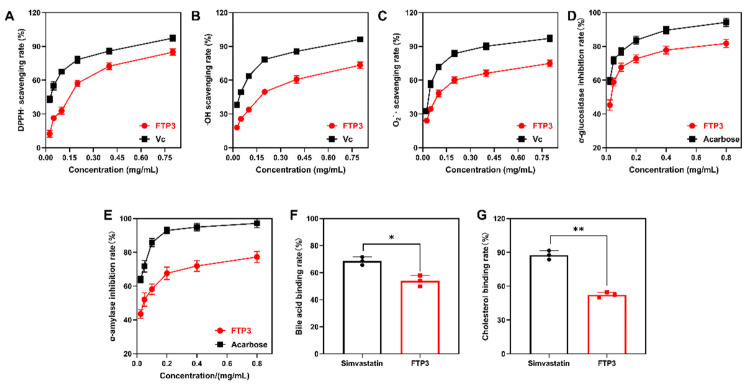
Antioxidant and glucolipid-metabolism-modulating activities of FTP3 in vitro. DPPH (**A**), hydroxyl (**B**), and superoxide anion (**C**) free radical scavenging rates. α-glucosidase (**D**) and α-amylase (**E**) inhibition rates. Bile acid (**F**) and cholesterol (**G**) binding rates. Compared with the simvastatin group, * *p* < 0.05, ** *p* < 0.01.

**Figure 3 foods-12-03428-f003:**
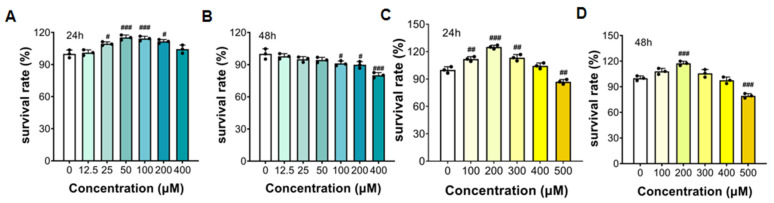
The survival rate of HepG2 cells with FTP3 and OA treatments. The survival rates of HepG2 cells with FTP3 for 24 h (**A**) and 48 h (**B**). The survival rates of HepG2 cells with OA for 24 h (**C**) and 48 h (**D**). # *p* < 0.05, ## *p* < 0.01, ### *p* < 0.001.

**Figure 4 foods-12-03428-f004:**
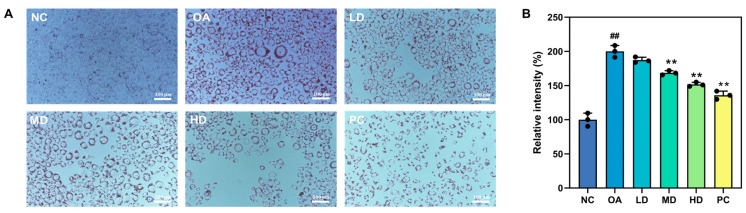
Oil Red O staining and lipid droplet content assay. (**A**) Intracellular lipid deposition in OA-induced HepG2 cells (Oil Red O staining, 200-fold). (**B**) The contents of intracellular lipid accumulation induced by oleic acid. Compared with the NC group, ## *p* < 0.01; compared with the OA group, ** *p* < 0.01. NC (negative control group). OA (oleic acid group, 400 μM OA). LD, MD, and HD (400 μM OA + 25, 50, 100 μg/mL FTP3). PC (positive control group, 100 μg/mL of simvastatin).

**Figure 5 foods-12-03428-f005:**
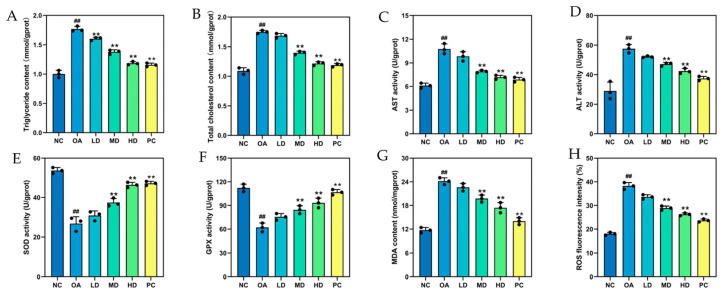
Lipid metabolism and antioxidant activity in OA-induced HepG2 cells. The contents of TG (**A**), TC (**B**), and MDA (**G**). The activities of AST (**C**), ALT (**D**), SOD (**E**), and GPX (**F**). ROS fluorescence intensity (**H**). Compared with the NC group, ## *p* < 0.01; compared with the OA group, ** *p* < 0.01.

**Figure 6 foods-12-03428-f006:**
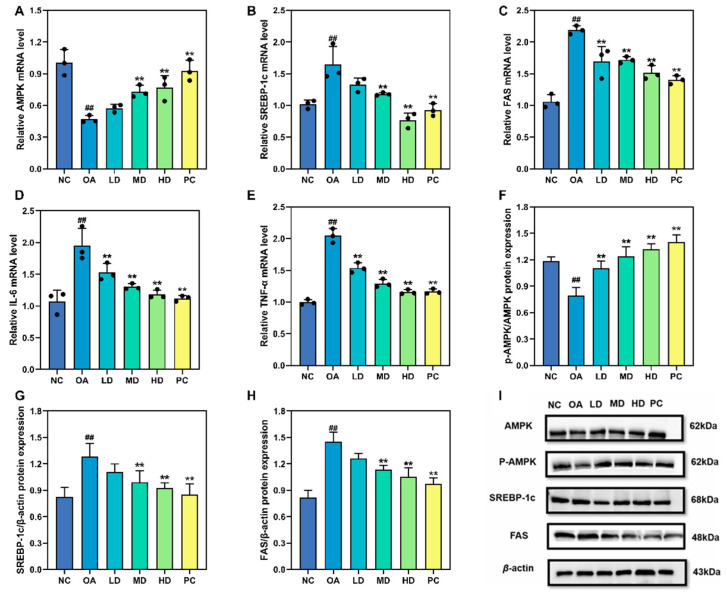
Gene and protein expression associated with lipid metabolism. mRNA levels of AMPK (**A**), SREBP-1c (**B**), FAS (**C**), IL-6 (**D**), and TNF-α (**E**). Protein expression of p-AMPK/AMPK (**F**), SREBP-1c/β-actin (**G**), and FAS/β-actin (**H**). Western blot protein expression bands (**I**). Compared with the NC group, ## *p* < 0.01; compared with the OA group, ** *p* < 0.01.

**Figure 7 foods-12-03428-f007:**
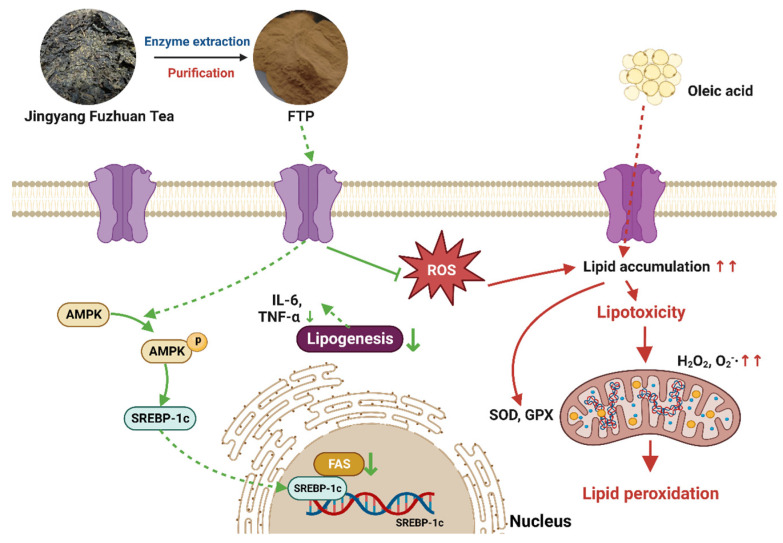
Mechanism of action of FTP3 in regulating lipid metabolism in HepG2 cells.

**Table 1 foods-12-03428-t001:** Sequences of primers used for quantitative real-time PCR.

Gene	Sequence (5′–3′)
SREBP-1c-F	GCGGAGCCATGGATTGCAC
SREBP-1c-R	CTCTTCCTTGATACCAGGCCC
FAS-F	TACATCGACTGCATCAGGCA
FAS-R	GATACTTTCCCGTCGCATAC
IL-6-F	ACTCACCTCTTCAGAACGAATTG
IL-6-R	CCATCTTTGGAAGGTTCAGGTTG
TNF-α-F	GAGGCCAAGCCCTGGTATG
TNF-α-R	CGGGCCGATTGATCTCAGC
β-actin-F	GCCGACAGGATGCAGAAGG
β-actin-R	TGGAAGGTGGACAGCGAGG

## Data Availability

The datasets generated for this study are available on request to the corresponding author.

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
