# Peer review of "Lipid-Lowering Effects of a Novel Polysaccharide Obtained from Fuzhuan Brick Tea In Vitro"

_foods, 2023, doi:10.3390/foods12183428_

Round 1
Reviewer 1 Report (Previous Reviewer 1)
The authors made efforts to deal with the points I made in the 1st round of revision and they have appropriately clarified some of them. It remains, however, that the language is poor, making the paper very difficult to follow. The limitation of absent evidence that concentrations as employed in vitro can be reached in tea drinkers still makes real life relevance of the findings doubtful.
The language in insufficient. This includes awkward wording, where one has to guess what the authors actually want to express, as well as specific grammar deficits (e.g. singular-plural).
Author Response
Please see the attachment.

Reviewer 2 Report (Previous Reviewer 2)
The abstract has been improved with the inclusion of a concise background section that provides context and rationale for the study. While the results were mentioned briefly previously, the authors have now included specific quantitative information. The extent of lipid accumulation reduction, changes in levels of TG, TC, ROS, MDA, ALT, AST, SOD, and GPX, as well as the upregulation of AMPK and downregulation of FAS, SREBP-1c, IL-6, and TNF-α genes and proteins, have been provided. The authors have made extensive work to improve the replicability of the study by improving the materials and methods section effectively.
Furthermore, it is essential to acknowledge that, apart from addressing all the mentioned comments and recommendations, the authors successfully demonstrated that FTP3 exhibits antioxidative and lipid-lowering properties by effectively reducing lipid accumulation in HepG2 cells. Through its modulation of key genes and proteins involved in glycolipid metabolism, FTP3 activates the AMPK signaling pathway, leading to a significant reduction in triglyceride and total cholesterol levels, as well as reactive oxygen species and malondialdehyde levels.
I commend the excellent quality of the English language used in the manuscript as it significantly enhances the paper's readability and communication. However, a few minor corrections related to formatting, referencing style, and grammar need attention. Please ensure consistent font usage, proper citation formatting, and review the document for any minor grammatical errors or typos.
Best regards,
Author Response
Please see the attachment.

Reviewer 3 Report (Previous Reviewer 3)
The questions are well answered and the quality of the manuscript has been significantly improved.
Author Response
Please see the attachment.

This manuscript is a resubmission of an earlier submission. The following is a list of the peer review reports and author responses from that submission.
Round 1
Reviewer 1 Report
In the first part of the paper (Figure 1), the authors describe in detail the chemical properties of FTP3. I am not a chemist and not competent on these methods. Hence, I shall not comment on this part of the study except for one point I do not understand: It is described in 3.1.1. that FTP3 contained different compounds, e.g. uronic acid. In line 418 the authors address the “composition” of FTP3 and its sheet shape with pores. But they also ascribe a specific molecular weight of 335.68 to FTP3. Is FTP3 a specific chemical entity, i.e. a compound with a specific molecular structure, or is it an extracted mixture of different molecules (e.g. of uronic acid and polysaccerides of different structure)?
The main problem of conclusions, which can be drawn from the biochemical and metabolic experiments, is that a highly artificial in vitro-approach was used (carcinoma cell line). The authors do neither provide evidence nor do they discuss, why their findings made in vitro could be of relevance under tea consumption in real life. More specifically: How did the authors select the concentrations of FTP3 used in the in vitro-experiments? Are these concentrations comparable to what possibly prevails in the intestine or in the blood of tea drinkers? Are relevant amounts of FTP3 lost by digestion in the intestine? Is FTP3 absorbed in unchanged form into the blood, is the liver of tea drinkers exposed to concentrations as used in the HepG2 cells? Figure 6 suggests that FTP3 is taken up into liver cells – is there any evidence that the compound reaches the liver and enters the cells in vivo?
Abbreviations used in the abstract should be defined at their first appearance. Abbreviations used in the main text should be defined at their first appearance in the main text (also, if already defined in the abstract). Abbreviations used in tables or figures should always be defined in the corresponding legends (also if already defined in the text).
The abbreviations used for the various groups shown in many figures are not defined (NC/OA/LD/MD/HD/PC). Reading the paper carefully, I may have correctly guessed what most of them stand for. But is impossible to show figures without clear information in the legend about what is shown.
I guess that OA stands for oleic acid. What was the concentration of oleic acid used in HepG2 cell experiments and how was oleic acid dissolved?
The authors repeatedly refer to “glycolipids”. It appears that this is not what they mean. Do they rather mean “lipids” or “lipid and glucose”?
Did the authors check the viability of the cells under the various conditions?
Caution is recommended in the interpretation of AMPK activation. AMPK activity is strongly affected by the cellular energy state (ATP/ADP/AMP concentration). The cellular energy state, and hence AMPK activity and downstream effects, is responsive to very unspecific stress factors in cultured cells.
Line 160: It appears that you are not testing the “lipid metabolism of FTP3” but rather the “effects of FTP3 on lipid metabolism”.
Line 275: Should probably be “cholesterol binding inhibition capacity”, not “cholesterol inhibition capacity”.
Line 302: AST and ALT as found in circulating plasma, not as prevailing in hepatocytes, are used as indicators of liver injury.
Line 367,368: “black tea polysaccharides have hypoglycemic activity to α-glucosidase and α-amylase”: Wording is inappropriate. I assume the authors want to state “hypoglycemic activity due to inhibition of the activities of α-glucosidase and α-amylase”.
Lines 384,385: “increase of TC and TG content may lead to intracellular fat accumulation and increase the risk of NAFLD”: Wording is inappropriate. The increase of TG content does not “lead to” but is the same as fat accumulation. And NAFLD is per definition an increase in liver TG.
Lines 427,428: The authors claim that they show “hypolipidemic” action. The term “hypolipidemic” refers to the lipid concentration in blood, but the authors did only in vitro-experiments, no measurements on blood. The experiments may hint at a possible hypolipidemic activity in vivo, but hypolipidemic activity is not shown in this paper.
Lines 429-432: The claim that FTP3 regulates lipid metabolism “by” AMPK/SREBP-1c/FAS pathway seems overshooting. All this is shown, but the assumption that the AMPK/SREBP-1c/FAS pathway is the cause for changes in lipid metabolism remains a speculation. This should be worded with more caution.
The paper contains a chemical characterization and several in vitro-tests on a compound (?) extracted from Fuzhuan black tea. There are several flawed statements and I cannot tell, to what extent this is due to language problems, lack of knowledge in content, or simple sloppiness. Methods lack information. I am hardly competent regarding the chemical part of the paper, but it remained unclear to me, whether the examined "FTP3" is a concrete chemical entity (i.e. a specific molecule with a defined structure) or an extracted mixture of different compounds. Also a major problem, the metabolic experiments have been done in highly artificial systems (carcinoma cell line) and it remains open, whether the experimental conditions (e.g. employed concentration of test compound) could in any way be relevant to under in vivo-conditions.
Reviewer 2 Report
Major revisions
(Since the materials and methods section lacks important details to assure the replicability and the reproducibility of the present study)
Report:
The paper provides an overview of the study on the lipid-lowering effects of Fuzhuan brick tea polysaccharide (FTP) and its newly isolated acidic variant, FTP3. However, several areas require improvement to enhance clarity and precision.
Abstract
The abstract could benefit from a concise background section to provide context and rationale for the study. This would involve briefly discussing the significance of elevated blood lipid levels and the current understanding of FTP's hypoglycemic and hypolipidemic effects.
· The results are mentioned briefly but lack specific quantitative information. It is essential to include specific data regarding the extent of lipid accumulation reduction, changes in levels of TG, TC, ROS, MDA, ALT, AST, SOD, and GPX, as well as the upregulation of AMPK and downregulation of FAS, SREBP-1c, IL-6, and TNF-α genes and proteins. This will help readers understand the significance and reliability of the findings.
· The abstract would benefit from a brief statement about the potential implications or applications of the study's findings. How could these results contribute to future research, clinical practice, or the development of therapeutic interventions?
Introduction
The introduction is well-written, no further comments.
Materials and methods
· In the "Isolation and Purification of Polysaccharides from FBT" subsection, mentioning the specific quantities or ratios of papain, cellulase, and FBT powder used during the extraction process would be helpful. Additionally, provide more details about the Sevag method deproteinization, concentration, precipitation, and lyophilization steps for obtaining crude FBT polysaccharide.
· In the "Isolation and Purification of Polysaccharides from FBT" subsection, it would be helpful to mention the specific quantities or ratios of papain, cellulase, and FBT powder used during the extraction process. Additionally, provide more details about the Sevag method deproteinization, concentration, precipitation, and lyophilization steps for obtaining crude FBT polysaccharide.
· When describing the purification of FTP using DEAE-cellulose DE-52 column chromatography and Sephadex G-200 columns, include the specific column dimensions and volumes used. This will assist in understanding the separation and purification process.
· Specify the NMR instrument's model and operating frequency used to analyze 1H-NMR and 13C-NMR spectra. Additionally, mention the solvent used and any sample preparation steps before measuring the NMR spectra.
· When describing the purification of FTP using DEAE-cellulose DE-52 column chromatography and Sephadex G-200 columns, include the specific column dimensions and volumes used. This will assist in understanding the separation and purification process.
· In the "Oil Red O staining and lipid droplets content assay" subsection, specify the staining protocol for Oil Red O, including the concentration and duration of staining. Additionally, provide more details about the digital camera and light microscope setup used for visualizing the stained lipid droplets. Mention any specific image analysis techniques employed for quantification.
In addition to the mentioned comments and suggestions, it is important to note that the discussion in the study is extensive, well-organized, and well-written. Furthermore, the results clearly indicate that FTP3 possesses antioxidative and lipid-lowering properties, as it effectively reduces lipid accumulation in HepG2 cells. By modulating key genes and proteins involved in glycolipid metabolism, FTP3 activates the AMPK signaling pathway, resulting in a notable decrease in triglyceride and total cholesterol levels, as well as reactive oxygen species and malondialdehyde levels. Additionally, it enhances the activities of antioxidant enzymes. These findings underscore the potential of FTP3 as a natural compound for addressing lipid-related disorders and emphasize the importance of exploring alternative strategies to mitigate the negative consequences of unhealthy lifestyle choices on human health.

The overall content and ideas are effectively conveyed. However, minor editing is needed to enhance the clarity and coherence of the writing. Specifically, attention should be given to sentence structure, grammar, punctuation, and word choice.
Reviewer 3 Report
This manuscript is dealing with a novel polysaccharide obtained from Fuzhuan brick tea alleviates glycolipid metabolism via the AMPK/SREBP-1c/FAS pathway. Generally, this manuscript has an interesting and practical subject. This manuscript needs some modifications.
Please choose better keywords. Keywords should not be present in the title.
The DPPH test method is not mentioned very well. The concentration of DPPH solution is very important and should be mentioned in the method section.
For SEM, samples were freeze-dried?
The AFM method needs more information, such as the type of the cantilever or the method of scanning.
The SEM figures have a poor quality and should be improved.
The conclusion part of the manuscript needs to be re-structured.
The reference style should be according to the journal guide. The references should be re-checked.
